# Physical networks as network-of-networks

Gábor Pete [1,2] ✉, Ádám Timár [1,3], Sigurdur Örn Stefánsson[3],
Ivan Bonamassa [4] & Márton Pósfai [4] ✉

Physical networks are made of nodes and links that are physical objects embedded in a geometric space. Understanding how the mutual volume exclusion between these elements affects the structure and function of physical networks calls for a suitable generalization of network theory. Here, we introduce a network-of-networks framework where we describe the shape of each extended physical node as a network embedded in space and these networks are bound together by physical links. Relying on this representation, we introduce a minimal model of network growth and we show for a general class of physical networks that volume exclusion induces heterogeneity in both node volume and degree, with the two becoming correlated. These emergent properties strongly affect the dynamics on physical networks: by calculating their Laplacian spectrum as a function of the coupling strength between the nodes we show that degree-volume correlations suppress the role of hubs as early spreaders in diffusive dynamics. We apply the network-of-networks framework to describe several real systems and find properties analog to the minimal model networks. The prevalence of these properties points towards general growth mechanisms that do not depend on the specifics of the systems.

The building blocks of physical networks are extended objects that do not intersect each other, resulting in non-trivial geometric layouts[1], link entanglement[2] and emergent correlations between physical and network structure[3]. Yet, these works model nodes as localized spheres connected by extended tube-like links, an assumption that does not necessarily reflect the structure of most real-world physical networks. In the connectome, for example, nodes represent neurons with non-trivial dendritic shapes, and links are point-like synapses[4]. A similar picture emerges for molecular networks such as the cytoskeleton, mitochondrial networks, or fiber materials, where nodes are extended molecular strands and bonds between them are localized[5–7], as well as in the wood-wide-web, where extended tree roots and mycelia connect to form a complex underground network[8,9]. Therefore, the sphere-tube paradigm often falls short of describing physical networks, calling for a more general framework to cope with the complex shape of nodes and links.

In this work, we develop a network-of-networks representation of physical networks that is able to capture arbitrary node shapes[10,11] and

allows us to characterize both structural and dynamical properties of networks. Relying on the network-of-networks framework, we introduce a model that grows physical networks from fractal segments. Analytically solving the model, we show that physicality induces heterogeneity in both the physical and the network properties of the nodes and that the two become strongly correlated. These correlations also affect the dynamics on the networks: generalizing the combinatorial Laplacian to physical networks[12–14], we show that fast dynamical modes associated to hubs (and corresponding to the tail of the Laplacian spectra) are suppressed by the emergent correlations between node volume and degree. The usefulness of the mathematical tools we develop in this paper goes beyond the specifics of the model, and we demonstrate this by applying our framework to several real physical networks, including a recently collected data set describing more than ~ 20, 000 neurons of the adult fruit fly's brain[15]. In doing so, we identify positive node degree-volume correlations similar to our minimal growth model, and we show that these have an analog effect on the Laplacian spectrum of the connectome. The fact that degree-

¹HUN-REN Alfréd Rényi Institute of Mathematics, Budapest, Hungary. ²Department of Stochastics, Institute of Mathematics, Budapest University of Technology and Economics, Műegyetem rkp. 3., H-1111 Budapest, Hungary. ³University of Iceland, Reykjavík, Iceland. ⁴Department of Network and Data Science, Central European University, Vienna, Austria. ✉e-mail: gabor.pete@renyi.hu; posfaim@ceu.edu

volume correlations emerge in a minimal model while also prevalent in real systems suggests a general mechanism behind such correlations that does not depend on the complex details of the growth of real networks.

## Results

### Network-of-networks representation

We aim to describe physical networks embedded in some substrate or medium. In its most general form, the substrate is represented by a graph $\mathcal{S}$, and each physical node $i$ is an extended object occupying a subgraph $\mathcal{V}_i \subset \mathcal{S}$. To capture volume exclusion, we do not allow nodes to overlap, i.e., $\mathcal{V}_i \cap \mathcal{V}_j = \varnothing$ for $i \neq j$. Two nodes $i$ and $j$ may form a link $(i,j)$ if they occupy adjacent sites in $\mathcal{S}$. The physical layout $\mathcal{P}$ of the network is a network-of-networks, i.e., it is the union of physical nodes, where each node is a network itself, together with the bonds forming the connections between the nodes (Fig. 1a). The layout $\mathcal{P}$ is a physical realization of the combinatorial network $\mathcal{G}$, where node $i$ of $\mathcal{G}$ corresponds to the physical node $\mathcal{V}_i$, and nodes $i$ and $j$ are connected if there is a bond between $\mathcal{V}_i$ and $\mathcal{V}_j$ in $\mathcal{P}$ (Fig. 1b). Though the substrate $\mathcal{S}$ can represent any available space, here we focus on substrates that are $d$ dimensional cubic lattices with linear size $L$ and periodic boundary conditions. Note that network representations of this kind are employed in the graph drawing literature with the focus on algorithms that embed a given combinatorial network into $\mathcal{S}$[16]. Here, we are interested in physical networks $\mathcal{P}$ growing in $\mathcal{S}$, the emergent relation between $\mathcal{P}$ and $\mathcal{G}$, and its consequences on the dynamics on the network.

### Network growth

To study the effect of physicality on network evolution, we define a model of network growth relying on the network-of-networks representation. We start with an empty $\mathcal{S}$ and we place a single physical node $\mathcal{V}_0$ occupying a subset of the sites. We add the rest of the nodes iteratively: At time step $t$ we add a new node $\mathcal{V}_t$ that is seeded at a random unoccupied site and grows until it hits an already existing node $\mathcal{V}_s$ and a link $(t-s)$ is formed. The growth of node $\mathcal{V}_t$ is driven by some random or deterministic process; and we assume that the physical nodes are characterized by a fractal dimension $d_f \in [1, d]$[17,18]. We add $N$ physical nodes or until all of $\mathcal{S}$ is occupied; in the latter case we call the physical network saturated.

Since the total volume of the network increases over time, later nodes hit the network at higher rates, and the typical size of nodes decreases. Hence, we expect that nodes added early have a higher degree than nodes added in the final stages of the network evolution, both because they are larger and they have more time to collect connections. This suggests that to analytically characterize the evolution

of the physical network two ingredients have to be considered: (i) network growth, i.e., nodes are added sequentially to the system and (ii) externally limited node growth, i.e., the nodes grow until they hit the already existing network. We show that these two ingredients lead to the emergence of power law combinatorial networks with degree exponents $\gamma \leq 3$.

We start the analytical treatment of the model by estimating the probability $p_{ij}$ that two randomly placed physical nodes $\mathcal{V}_i$ and $\mathcal{V}_j$ intersect. If the two boxes containing the physical nodes have side length $l_i \gg l_j$, respectively, and the larger node $\mathcal{V}_i$ intersects the box containing the smaller node $\mathcal{V}_j$, then, by dimension count, the two nodes overlap with positive probability if $d_f \geq d/2$. We can tile the lattice with $(L/l_j)^d$ boxes with side length $l_j$, and the number of such boxes intersected by $\mathcal{V}_i$ is $\sim (l_i/l_j)^{d_f}$. Therefore the intersection probability is

$$p_{ij} \sim \frac{l_i^{d_f} l_j^{d-d_f}}{L^d} \sim \frac{v_i v_j^{d/d_f - 1}}{L^d}, \tag{1}$$

where $v_i = |\mathcal{V}_i| \sim l_i^{d_f}$ is the volume of node $i$. If, however, $d_f < d/2$ and $l \lesssim l_i \ll L$, then the nodes avoid each other with high probability. In this case, the intersection probability will have the meanfield behavior, well-approximated by the probability of selecting the sites of $\mathcal{V}_i$ and $\mathcal{V}_j$ uniformly from $\mathcal{S}$, i.e., $p_{ij}^{\text{MF}} \sim v_i v_j / L^d$, which is independent of $d_f$ and agrees with Eq. (1) for $d_f = d/2$.

Using the same box-counting argument that led to Eq. (1), the probability that a node added at time $t$ intersects any existing node $s < t$ is approximately given by $\sum_{s<t} p_{st} = v_t^{d/d_f} V_{t-1}/L^d$, where $V_{t-1}$ is the total volume of nodes $s < t$. A key observation is that the size of node $t$ increases until it hits the existing network, meaning that $v_t$ increases until $\sum_{s<t} p_{st} \approx 1$, allowing us to estimate the volume of node $t$ as

$$v_t \approx \left( V_{t-1}/L^d \right)^{-\frac{d_f}{d-d_f}}. \tag{2}$$

Equation (2) allows us to express the evolution of the expected total volume via the recursion $V_{t+1} = v_{t+1} + V_t$ with initial condition $V_0 = v_0$. Using a continuous time approximation, we obtain

$$V_t \approx L^d \left[ \frac{d}{d-d_f} \frac{t+c}{L^d} \right]^{\frac{d-d_f}{d}} \sim L^d \left( \frac{t}{L^d} \right)^{1-d_f/d}, \tag{3}$$

where $c$ is a constant depending on $v_0$. A natural choice for the latter is that the first node spans the entire available space, i.e., $v_0 \sim L^{d_f}$, in which case $c$ is independent of $L$. Equation (3) predicts that $N_{\text{sat}}$, the number of nodes when the network saturates, scales as $N_{\text{sat}} \sim L^d$, meaning that the average node volume $\langle v \rangle$ remains constant in the $L \to \infty$ large system limit. Therefore, the physical layout $\mathcal{P}$ is optimal in the sense that no physical representation of a combinatorial network of $N_{\text{sat}}$ nodes can fit into a smaller volume than $\sim N_{\text{sat}} \sim L^d$. It is noteworthy that the model achieves this bound despite the fact that the nodes grow randomly.

In light of Eq. (3), we can now calculate the expected degree of the physical nodes in the combinatorial network $\mathcal{G}$. In the continuous time approximation, the volume of the newly added node $v_t$ is provided by the time derivative of $V_t$, i.e., $v_t \sim \left( t/L^d \right)^{-d_f/d}$. Hence, following Eq. (1), the expected degree of node $t$ after the addition of $N$ nodes is

$$k_t(N) = 1 + \frac{v_t}{L^d} \sum_{s=t+1}^{N} v_s^{d/d_f - 1} \sim v_t \cdot \left( \frac{N}{L^d} \right)^{\frac{d}{d_f}}, \tag{4}$$

where the proportionality is valid for $t \ll N$. This means that the volume occupied by large nodes (i.e., nodes that were added early) in the physical layout is proportional to their degree in the combinatorial network.

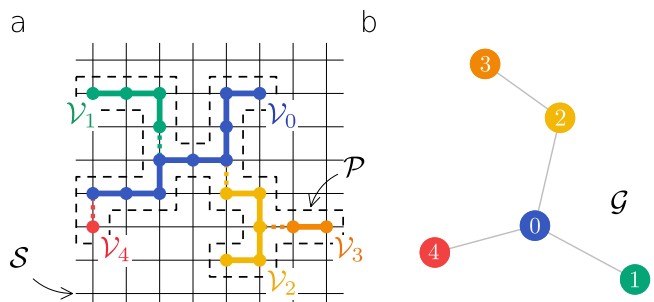

**Fig. 1 | Network-of-networks representation. a** Each physical node $\mathcal{V}_i$ is a subgraph of the substrate $\mathcal{S}$. Physical nodes cannot overlap, i.e., $\mathcal{V}_i \cap \mathcal{V}_j = \varnothing$ for $i \neq j$. The physical layout $\mathcal{P}$ (dashed area) is a network-of-networks: it is the union of physical nodes $\mathcal{V}_i$ together with the bonds connecting them. **b** The combinatorial network $\mathcal{G}$ is a coarse-grained representation of $\mathcal{P}$ capturing the connections between the nodes without the physical structure.

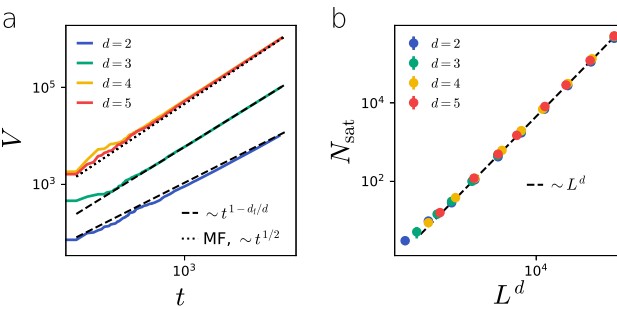
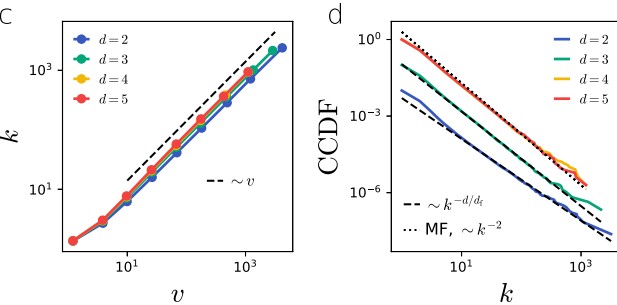

**Fig. 2 | Evolution of LERW physical networks. a** The temporal evolution of the total volume $V_t$ of physical networks, dashed lines represent the theoretical prediction Eq. (3). Networks built from LERWs in dimensions $d≥4$ fall into the mean-field regime. **b** The number of physical nodes in the saturated networks is proportional to the volume of the substrate $|\mathcal{S}|$ irrespective of $d_f$ and $d$. **c** Node degree is proportional to node volume independently from $d_f$ and $d$. **d** The complementary cumulative degree distribution function (CCDF) of the physical

networks. Dashed lines indicate the predicted degree exponent $γ = 1 + d/d_f ≤ 3$, while the dotted line corresponds to the mean-field behaviour, $γ_{MF} = 3$. Plots (**a**), (**c**) and (**d**) represent measurements of single networks with initial condition $v_0 = L^{d_f}$ and $|\mathcal{S}| = 10^6$. In (**b**) markers are an average of 10 independent networks, error bars represent the standard error of the mean. Lines corresponding to different slopes are shifted to increase readability.

We finally calculate the complementary cumulative degree distribution $P(k) = 1 - \frac{1}{N}\sum_{t:k_t ≥ k}^{N} 1$, finding that $P(k) \sim k^{-(γ-1)}$ with exponent

$$γ = 1 + \frac{d}{d_f}. \quad (5)$$

For $d_f ≤ d < 2d_f$ the degree exponent falls in the range $2 ≤ γ < 3$. In the mean-field regime, the degree exponent can be obtained by substituting $d/d_f$ with 2, yielding $γ_{MF} = 3$. Note that the upper critical dimension of the physical network depends on the kinetic growth of the nodes. For example, growing nodes along a straight trajectory in a random direction generates nodes with $d_f = 1$; therefore, the networks fall in the mean-field regime $d_f ≤ d/2$ even for embedding dimension $d = 2$.

In the above model, each physical node grows starting from a random location following some growth process. We stress that our calculations hold for a general class of node growth algorithms, the crucial assumption being that if the boxes around two random walk pieces intersect, then with uniformly positive probability the trajectories also intersect, which implies a level of isotropy of node growth. As a counterexample, consider nodes that always grow in one direction along one of the axes. Such nodes will run parallel to each other, avoiding intersection, hence the resulting network will be a collection of disconnected chains. If, however, the nodes grow along straight lines but in random directions, thus restoring isotropy on average, then the resulting network has a power law degree distribution (SI Sec. S1.4).

## Numerical simulations

To test our analytical predictions, we numerically generate physical networks where nodes grow according to random walk trajectories. Specifically, we generate nodes using loop-erased random walks (LERWs), i.e., a trajectory that evolves as a simple random walk in which any loop is erased as soon as it is formed[19–22]. Here, we focus on the LERW, as it represents a tractable model of non-self-intersecting random trajectories with well-understood non-trivial critical properties. Its critical properties are studied both in the mathematics and physics literature[23–26]; for example, their fractal dimension in $d = 2$ is $d_f = 5/4$[22], in $d = 3$ it is $d_f ≃ 1.6236(4)$[27,28], while its upper critical dimension is $d_u = 4$ where $d_f = 2$ with a logarithmic correction[29]. (See "METHODS" for further details.) We remark that our predictions are not specific to LERWs, in Sec. S1 of the Supplementary Information, we study various alternative growth processes.

Knowing the fractal dimensions of LERWs allows us to directly verify the predictions of Eqs. (3)–(5):

**Volume evolution.** Equation (3) predicts that the total volume of the physical network evolves as $V_t \sim t^{1-d_f/d}$. Figure 2a shows the excellent agreement between the theoretical predictions and numerical simulations. Note that, as expected, in the mean-field regime $d≥4$ the network volume follows the classic diffusion growth $V_t \sim t^{1/2}$. Figure 2b further corroborates the predicted scaling of the number of nodes at saturation, i.e., $N_{sat} \sim L^d$.

**Degree-volume correlations.** A second prediction is the emergence of degree-volume correlations, capturing the interplay between the physical layout $\mathcal{P}$ and the combinatorial network $\mathcal{G}$. In particular, Eq. (4) predicts a linear proportionality between the node volume $v_i$ and degree $k_i$, and we again find excellent agreement with simulations for all the tested dimensions (Fig. 2c).

**Power law emergence.** As a final test, we verify the emergence of power law scaling in the degree distribution of the combinatorial networks $\mathcal{G}$. As anticipated in Eq. (5), the degree exponent depends on both the dimensionality of the embedding substrate, $d$, and the fractal dimension of the nodes, $d_f$. Figure 2d shows that numerical simulations confirm the predicted degree exponent $γ = 1 + d/d_f$ for dimensions $d < 4$, while the mean-field exponent $γ_{MF} = 3$ is found for $d≥4$. In traditional models of combinatorial networks, heterogeneity typically arises from preferential attachment or some other optimization process. Our model is based on random growth without any explicit preference to create highly connected nodes; therefore, the uniform attachment tree may be considered as the non-physical counterpart of our model. Uniform attachment yields exponential degree distribution, hence the power law distribution observed here is a direct consequence of volume exclusion, which, together with the dynamic network growth, induces effective preferential attachment.

## Physical network Laplacian

The layout $\mathcal{P}$ is a physical realization of the combinatorial network $\mathcal{G}$. Traditional studies of dynamics on physical networks ignore the layout $\mathcal{P}$ and focus only on the role of $\mathcal{G}$, thus prompting the question: does modeling dynamics on $\mathcal{G}$ accurately capture dynamics on physical networks? To explore this, we study the spectral properties of $\mathcal{P}$ and show that physical nodes emerge as functional units through timescale separation, yet even in this limit the structure of $\mathcal{P}$ continues to affect the dynamics. We focus on the Laplacian spectrum[12] since it influences the behavior of several dynamical processes on networks[30] including diffusion[31,32], synchronization[33] and it underlies the definition of several information-theoretic tools to analyze the multiscale functioning of networks[10,14,34–37].

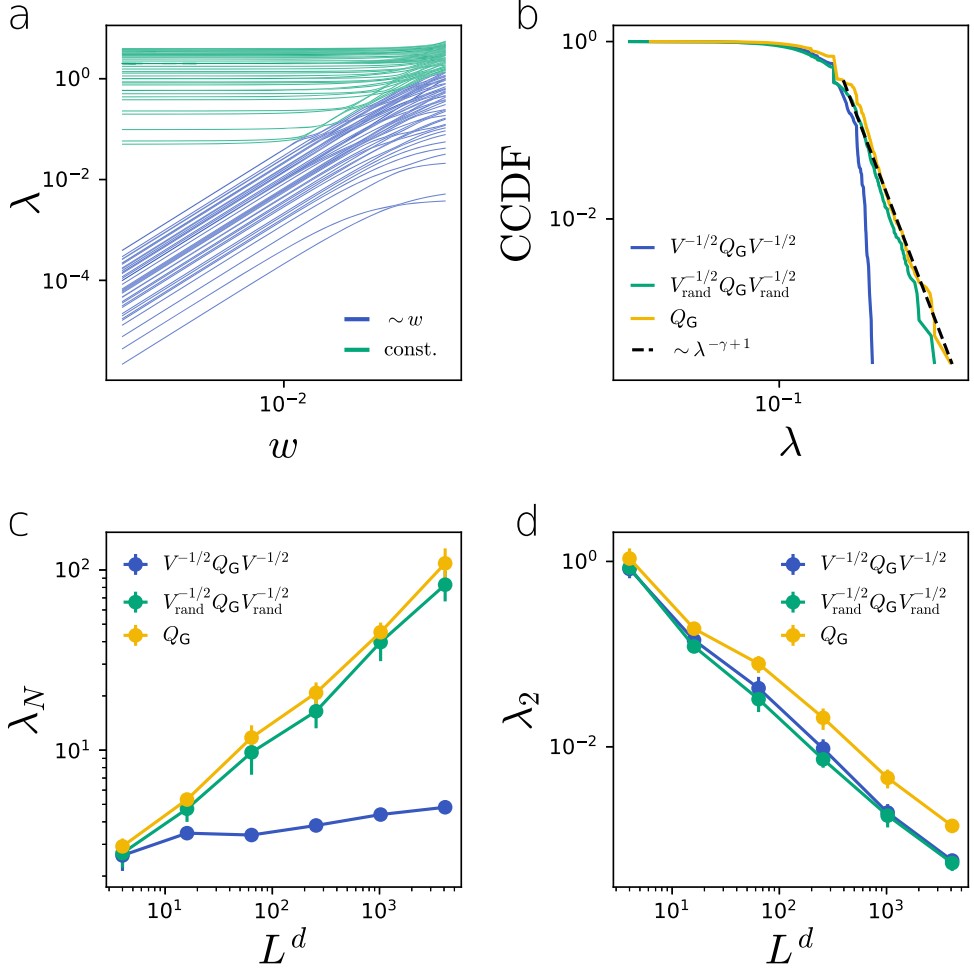

**Fig. 3 | Laplacian of LERW physical networks. a** For decreasing weight $w$ the eigenvalues of $\mathbf{Q}_{\mathcal{P}}$ separate into two groups: eigenvalues corresponding to the zero eigenmodes of $\mathbf{Q}_{\mathcal{P}}(w=0)$ decay as $\sim w$ (blue), while the rest converge to a constant value (teal). (**b**–**d**) Comparing the spectrum of the physical Laplacian $\mathbf{Q}_{\text{phys}} = \mathbf{V}^{-1/2}\mathbf{Q}_{\mathcal{G}}\mathbf{V}^{-1/2}$, the randomized physical Laplacian $\mathbf{Q}_{\text{phys}}^{\text{rand}} = \mathbf{V}_{\text{rand}}^{-1/2}\mathbf{Q}_{\mathcal{G}}\mathbf{V}_{\text{rand}}^{-1/2}$, and the Laplacian of the combinatorial network $\mathbf{Q}_{\mathcal{G}}$. **b**, **c** The heterogeneous node volume distribution and the correlation between node degree and volume significantly reduce the largest eigenvalues of the spectra. **d** Heterogeneous node volumes alone explain the reduction of the algebraic connectivity $\lambda_2$. Eigenvalues are calculated for $d = 2$ and $L = 10$ in (a) and $L = 100$ in (b). In (c) and (d) markers are an average of 10 independent networks, error bars represent the standard error of the mean.

We study the problem by invoking, once again, the network-of-networks representation. In our setup, we assign a weight to each connection in $\mathcal{P}$ such that links within physical nodes have weight 1 and links connecting two physical nodes have weight $w$, capturing that in real physical networks bonds between nodes are often qualitatively different than those within nodes. The weighted Laplacian matrix of $\mathcal{P}$ occupying $V$ sites is then $\mathbf{Q}_{\mathcal{P}} = \mathbf{D}_{\mathcal{P}} - \mathbf{A}_{\mathcal{P}}$, where $\mathbf{A}_{\mathcal{P}}$ is the $V \times V$ weighted adjacency matrix and $\mathbf{D}_{\mathcal{P}}$ is a diagonal matrix such that $[\mathbf{D}_{\mathcal{P}}]_{ss} = \sum_u [\mathbf{A}_{\mathcal{P}}]_{su}$ is the sum of the weights of the links adjacent to site $s$ in $\mathcal{P}$. If we now set $w = 0$, the network-of-networks falls apart and each physical node becomes a separate connected component, resulting in a block-diagonal Laplacian $\mathbf{Q}_{\mathcal{P}}(0) = \text{diag}(\mathbf{Q}_{\mathcal{V}_1}, \mathbf{Q}_{\mathcal{V}_2}, \ldots, \mathbf{Q}_{\mathcal{V}_N})$, where $\mathbf{Q}_{\mathcal{V}_i}$ is the Laplacian of the physical node $\mathcal{V}_i$. The Laplacian $\mathbf{Q}_{\mathcal{P}}(0)$ has $N$ zero eigenvalues corresponding to the $N$ blocks (i.e., the physical nodes), hence we can assign an eigenvector $\mathbf{u}_i(w=0)$ to the $i$-th node such that $[\mathbf{u}_i(0)]_s = 1/\sqrt{v_i}$ if site $s$ is within node $i$, otherwise $[\mathbf{u}_i(0)]_s = 0$, where $v_i = |\mathcal{V}_i|$ is the volume of node $i$. Since linear combinations of these vectors are also eigenvectors, we can write the zero eigenvectors of $\mathbf{Q}_{\mathcal{P}}$ as $\mathbf{u}(0) = \mathbf{M}\tilde{\mathbf{u}}$, where $\mathbf{M}$ is the $N \times V$ membership matrix such that $[\mathbf{M}]_{si} = 1/\sqrt{v_i}$ if site $s$ is part of node $i$, otherwise $[\mathbf{M}]_{si} = 0$, and $\tilde{\mathbf{u}}$ is any normalized $N$ dimensional vector.

We can gain insights about the spectral properties of $\mathbf{Q}_{\mathcal{P}}$ by working in the weak coupling regime $w \ll 1$ and relying on perturbation theory. Following a treatment similar to the one adopted to study diffusion in multilayer networks[38–40], we consider $w$ a small perturbation and write $\mathbf{Q}_{\mathcal{P}}(w) = \mathbf{Q}_{\mathcal{P}}(0) + w\mathbf{Q}'_{\mathcal{P}}$, where $\mathbf{Q}'_{\mathcal{P}}$ is the Laplacian matrix of the subnetwork of $\mathcal{P}$ formed by the bonds between physical nodes. The characteristic equation, up to first order in $w$, becomes then

$$
\begin{aligned}
&(\mathbf{Q}_{\mathcal{P}}(0) + w\mathbf{Q}'_{\mathcal{P}})(\mathbf{u}(0) + w\mathbf{u}') \\
&\approx (\lambda(0) + w\lambda')(\mathbf{u}(0) + w\mathbf{u}').
\end{aligned}
\tag{6}
$$

Perturbations around $\lambda(0) = 0$ lead to $N$ eigenvalues that are $\mathcal{O}(w)$, while the rest of the eigenvalues are constant in leading order (Fig. 3a). This means that on the $1/w$ timescale, diffusion-like dynamics on the physical network are captured by the $N$ slow eigenmodes. We obtain these from Eq. (6) (see "METHODS"), yielding

$$
\mathbf{V}^{-1/2}\mathbf{Q}_{\mathcal{G}}\mathbf{V}^{-1/2}\tilde{\mathbf{u}} = \lambda'\tilde{\mathbf{u}},
\tag{7}
$$

where $\mathbf{Q}_{\mathcal{G}}$ is the $N \times N$ Laplacian matrix of the combinatorial network $\mathcal{G}$ and $\mathbf{V}$ is an $N \times N$ diagonal matrix such that its diagonal elements are $[\mathbf{V}]_{ii} = v_i$. We call the volume-normalized Laplacian the physical network Laplacian $\mathbf{Q}_{\text{phys}} = \mathbf{V}^{-1/2}\mathbf{Q}_{\mathcal{G}}\mathbf{V}^{-1/2}$.

Equation (7) is a key relation for understanding the dynamics on physical networks since it allows to characterize the dynamics on $\mathcal{P}$ on

the timescale $1/w$ in a coarse-grained way: after integrating out the fast modes corresponding to eigenvalues $\lambda(w) \gg w$, the state of each physical node $\mathcal{V}_i$ is given by a single variable, while the coupling between the nodes is provided by the combinatorial network $\mathcal{G}$. However, the combinatorial Laplacian $\mathbf{Q}_\mathcal{G}$ is not sufficient to capture the dynamics, and we must normalize $\mathbf{Q}_\mathcal{G}$ by the volume of the nodes, as shown in Eq. (7). This means that physical networks with the same combinatorial network but different layout can have drastically different dynamical properties. For example, if nodes have approximately the same size, i.e., $v_i \approx V/N$, then the physical layout only affects the overall timescale, otherwise the Laplacian spectrum is determined by $\mathbf{Q}_\mathcal{G}$. If, however, node sizes are heterogeneously distributed, normalizing by volume will also have a heterogeneous effect on the eigenvalues.

**Application to the physical network growth model.** We showed above that physical networks generated by our network growth model are characterized by heterogeneous node-volume distribution and proportionality between the degree and the volume of nodes (Fig. 2). To probe the effect of this emergent correlation, we shuffle the volume of the nodes of a LERW physical network to remove the correlation between network and physical structure. We then compare the spectrum of the volume-normalized Laplacian $\mathbf{Q}_{\mathrm{phys}} = \mathbf{V}^{-1/2}\mathbf{Q}_\mathcal{G}\mathbf{V}^{-1/2}$ to its randomized version $\mathbf{Q}_{\mathrm{phys}}^{\mathrm{rand}} = \mathbf{V}_{\mathrm{rand}}^{-1/2}\mathbf{Q}_\mathcal{G}\mathbf{V}_{\mathrm{rand}}^{-1/2}$ and to the Laplacian spectrum of the combinatorial network $\mathbf{Q}_\mathcal{G}$. Figure 3b shows that the spectrum of $\mathbf{Q}_\mathcal{G}$ has a heavy tail characterized by the same $\gamma$ exponent of Eq. (5), as expected for combinatorial networks with power law degree distributions[12]. Adding heterogeneous but uncorrelated node sizes does not influence the tail while taking into account the degree-volume correlation of nodes removes the heavy tail and leads to a rapidly decaying spectrum. In power law networks, the eigenvector corresponding to the largest eigenvalue $\lambda_N$ of $\mathbf{Q}_\mathcal{G}$ is typically concentrated on the node with the largest degree[41,42]. In our model, the largest degree node also has the largest volume; therefore normalizing by node volume $\mathbf{V}^{-1/2}\mathbf{Q}_\mathcal{G}\mathbf{V}^{-1/2}$ significantly lowers $\lambda_N$. Since node sizes are heterogeneously distributed, with high probability, we associate volume ~ 1 to the highest degree node after randomization. Hence, the eigenvalue $\lambda_N$ of $\mathbf{Q}_\mathcal{G}$ is largely unaffected by the randomized normalization (Fig. 3c). At the other end of the spectrum, controlling the long-time mixing of the dynamics, the eigenvector associated with the algebraic connectivity $\lambda_2$ typically spans the entire network. Figure 3d shows that taking node volumes into account slows the dynamics down; however, degree-volume correlations do not significantly affect $\lambda_2$.

Note that positive degree-volume correlations, responsible for the suppression of the tail of the Laplacian spectrum, naturally arise in minimum-volume physical realizations of combinatorial networks. Any combinatorial network $\mathcal{G}$ has many possible physical realizations $\mathcal{P}$, a minimum volume realization is a $\mathcal{P}$ that minimizes the total volume of the network. Consider node $i \in \mathcal{G}$ with degree $k_i$; in any possible $\mathcal{P}$, the physical realization of node $i$ must have volume at least proportional to $k_i$, otherwise it is unable to support $k_i$ connections. Therefore, we expect positive degree-volume correlations in minimum-volume physical layouts. This means that any physical network generation process that minimizes total volume – either explicitly or as an emergent property, like in our model – is characterized by positive degree-volume correlations and hence that the spectrum of $\mathbf{Q}_{\mathrm{phys}}$ is similarly affected by physicality as in our model.

## Real physical networks

We identified the degree-volume correlations and the profile of the Laplacian spectrum as important features of physical networks that can emerge even in the simplest models. To measure these properties, we do not need a detailed description of the layout of a physical system – we only need the combinatorial network and a list of the node volumes, allowing us to describe very large and complex physical networks. As a case study, we investigate a recently published data set providing the three-dimensional layout of more than 20,000 neurons of the brain of an adult fruit fly and the location of more than 13 million synapses connecting them (Fig. 4a)[15]. Although our simple growth model does not attempt to capture the myriad of complex mechanisms shaping brain development, we find that the fruit fly brain is characterized by similar emergent properties as the model networks. Figure 4b shows, for example, that the multiplicity-weighted node degree, i.e., the number of synapses a neuron has, can be approximated by a power law $\gamma_{\mathrm{ff}} \approx 2.3$, albeit with an exponential cutoff[43,44]. We also find a strong positive correlation between the weighted degree and the volume of the nodes (Fig. 4c).

To compare the spectrum of the combinatorial Laplacian $\mathbf{Q}_\mathcal{G}$ and the physical network Laplacian $\mathbf{Q}_{\mathrm{phys}} = \mathbf{V}^{-1/2}\mathbf{Q}_\mathcal{G}\mathbf{V}^{-1/2}$, we measure volume in units such that the mean node volume is unity, i.e., $\langle v \rangle = 1$ (see "METHODS" for further details). Calculating the leading eigenvalues of $\mathbf{Q}_\mathcal{G}$ and $\mathbf{Q}_{\mathrm{phys}}$, we find that $\lambda_N^\mathcal{G}/\lambda_N^{\mathrm{phys}} \approx 32.7$, indicating that degree-volume correlations greatly suppress the modes of the dynamics that spread the fastest, similarly to model networks. This is further supported by Fig. 4d, showing again that physicality suppresses the tail of the spectrum.

To further probe the role of degree-volume correlations, we calculate the leading eigenvectors $\tilde{\mathbf{u}}_N$ of $\mathbf{Q}_\mathcal{G}$ and $\mathbf{Q}_{\mathrm{phys}}$. Figure 4e, g show that, as expected for heterogeneous combinatorial networks, $\tilde{\mathbf{u}}_N^\mathcal{G}$ is concentrated on the largest hub $i_\mathcal{G}$ in the network, and the weight of the eigenvector decays exponentially as the geodesic distance from $i_\mathcal{G}$ in $\mathcal{G}$. This means that, without taking physicality into account, the largest degree node is also the earliest spreader of diffusive dynamics. For the physical Laplacian $\mathbf{Q}_{\mathrm{phys}}$ we find a different picture: $\tilde{\mathbf{u}}_N^{\mathrm{phys}}$ is again concentrated on a single node $i_{\mathrm{phys}}$; this node, however, is not the largest hub. The leading eigenvector instead is centered on a node that balances high degree and low volume: node $i_{\mathrm{phys}}$ is the 159th largest degree node and is at the top 15 percentile of the volume distribution. In fact, node $i_{\mathrm{phys}}$ is the node that maximizes the degree-volume ratio, i.e., $i_{\mathrm{phys}} = \mathrm{argmax}_i k_i/v_i$. This means that degree-volume correlations not only slow down spreading dynamics in physical networks, but also change the identity of the early spreaders.

Here we chose to focus on the fruit fly brain network as it represents one of the largest and most detailed maps of physical networks available; however, our framework is not specific to neural networks. In the Sec. S2 of the Supplementary Information, we analyze four additional real systems: a network describing the cavities of a porous material, a neural network of a nematode, a river network, and a vascular network. In each case, we find positive degree-volume correlations and that these correlations suppress the tail of the Laplacian spectra. The fact that the physical and network properties of nodes become intertwined in such a diverse set of real networks, and also in the simplest models, indicates a general mechanism behind the emergence of degree-volume correlations that do not depend on the details of the individual networks.

## Discussion

Physical networks are complex networks that have a complex three-dimensional layout. The network-of-networks framework naturally lends itself to representing these systems: representing nodes as physically embedded networks allows us to capture arbitrary node shapes and complex wiring. Here, we relied on the network-of-networks framework to characterize both model and real physical networks. We identified correlations between node degree and volume as a prevalent feature of physical networks: We analytically showed that degree-volume correlations emerge in a minimal network growth model, in fact, we provided arguments that such correlations naturally arise through any growth process that minimizes network volume. We also showed that positive degree-volume correlations are generally present in real systems. These correlations have important

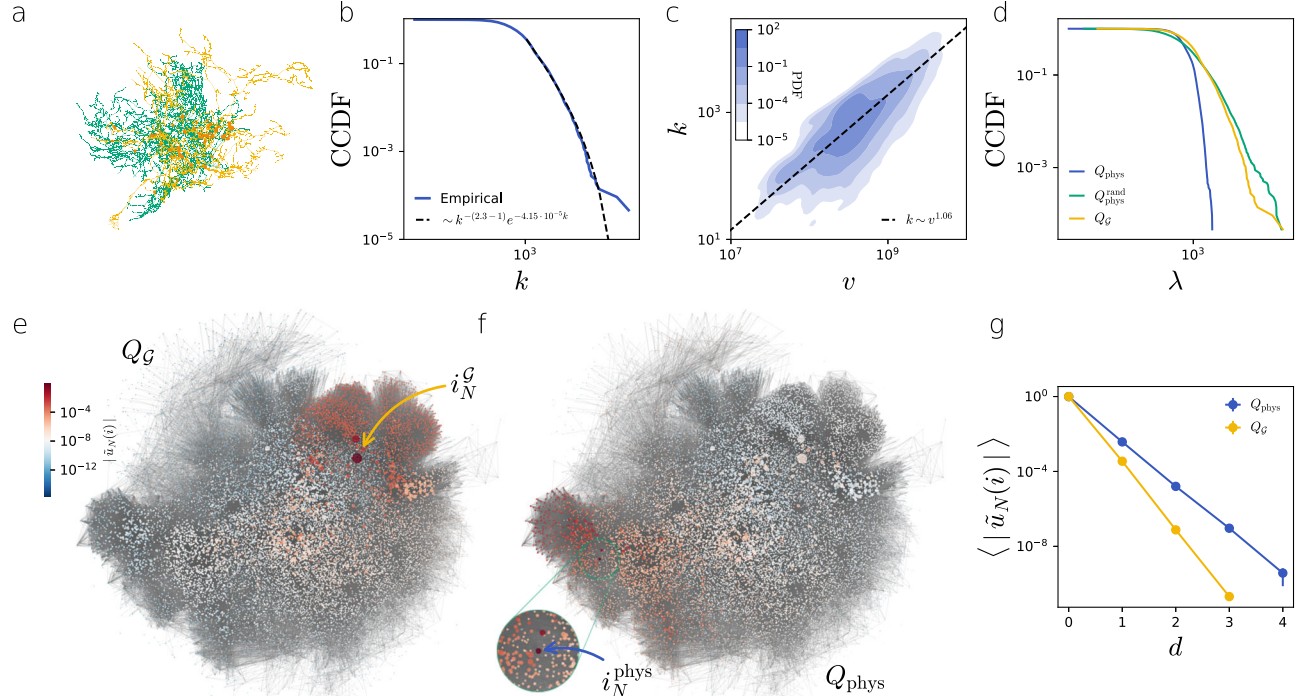

**Fig. 4 | Fruit fly brain network. a** The neurons have complex three-dimensional shapes. Two intertwined neurons (teal and yellow) are connected by synapses (red circles). **b** The tail of the weighted degree distribution can be approximated with the power law with $\gamma_{ff} \approx 2.3$ with an exponential cutoff. **c** Similar to the network growth model, the volume of the nodes $v$ is strongly correlated with their degree $k$. The markers indicate binned degree-volume averages, where the data points are binned based on node volume. The shading represents a kernel density estimate of the joint $v$-$k$ distribution, the dashed line indicates the least squares fit of the power law scaling. **d** The effect of the node degree-volume correlation on the Laplacian spectrum in the brain network is analog to the effect of correlations in the network growth model. The spectra of $\mathbf{Q}_{phys}$ and $\mathbf{Q}_{phys}^{rand}$ are shifted to the right to allow direct comparison of the tails of the distributions. **e**, **f** A visualization of the leading eigenvectors $\tilde{\mathbf{u}}_N$ of the combinatorial and physical Laplacians. The color of each node $i$ corresponds to $|\tilde{u}_N(i)|$, the weight of the leading eigenvector at node $i$, and the size of the nodes is a linear function of their degree. (e) The eigenvector $\tilde{\mathbf{u}}_N^{\mathcal{G}}$ is concentrated on the node with the highest degree $i_{\mathcal{G}}$. **f** Due to degree-volume correlations, the $\tilde{\mathbf{u}}_N^{phys}$ is concentrated on node $i_{phys}$, which has the maximum degree-volume ratio in the network. **g** The weight of the leading eigenvectors $|\tilde{u}_N^{\mathcal{G}}(i)|$ and $|\tilde{u}_N^{phys}(i)|$ decays exponentially as a function of the geodesic distance from $i_{\mathcal{G}}$ and $i_{phys}$, respectively. Error bars indicate the standard error of the mean and are typically smaller than the marker size. Node locations in (**e**) and (**f**) are generated based on $\mathcal{G}$, and do not correspond to the actual physical locations.

consequences on dynamics unfolding on physical networks: the tail of the physical Laplacian spectrum is suppressed by the large volume of hubs. More broadly, these results vividly demonstrate that traditional methods of network science focusing on combinatorial networks cannot fully describe physical networks and that their three-dimensional layout must be accounted for.

Our work opens new avenues for physical network research in several ways. First, by establishing the connection between physical networks and network-of-networks, we allow future work to leverage the rich literature of multi-layer networks to characterize physical systems[10,11]. For example, multi-layer centrality measures can be used to quantify the importance of physical nodes[45–47]. Second, previous work on physical networks relies on methods that require a full description of their spatial layout and, therefore, are often limited to systems of a few hundred nodes[1–3]. In contrast, the quantities we studied can be measured relying on the combinatorial network and a list of node volumes, allowing the characterization of large-scale physical networks without the need of the full three-dimensional layout. For example, we can tune the volume of the nodes to systematically study how physical layout affects the Laplacian spectrum. Finally, the simple growth model and its analytical description can serve as the starting point for the exploration of additional growth mechanisms that characterize neural networks and other physical networks. For example, future work may study branched nodes, long-range interactions that guide the growth of physical nodes, or the expansion of available space by modeling the evolution of the underlying substrate.

## Methods

### Loop-erased random walks

In our network growth model, we can generate physical nodes with any stochastic or deterministic process that produces a growing fractal embedded in $\mathbb{Z}^d$. Standard self-avoiding walks are traditionally used to model polymers obeying volume exclusion and, therefore, represent a natural choice to model node growth[17]. However, the naïve kinetic version of the self-avoiding walk traps itself in two and three dimensions at finite length[21], making it a poor candidate for constructing large physical networks. Instead, we focus on loop-erased random walks (LERW): a LERW evolves as a simple random walk, except when it intersects itself, we delete the loop that it created and continue the walk[20]. This guarantees that the final physical node does not intersect itself and that the walk never gets trapped. Alternatively, the LERW can be defined as a special case of Laplacian-random walks, where transition probabilities are defined by a harmonic function[48,49]. This alternative construction does not require deleting loops, hence is more realistic as a growth model. The LERW has attractive mathematical properties making it amenable to analytical treatment. For example, Wilson's algorithm uses iterative LERWs to construct a uniform spanning tree (UST) of any graph[24]. In fact, the physical network our algorithm constructs is a UST of the $\mathcal{S}$ substrate together with a partition identifying the nodes. Future work may exploit this connection between USTs and LERW physical networks, together with known results in dimensions $d = 2$ and $d > 4$[22,50], to rigorously prove some of the results presented here.

## Perturbation of the physical Laplacian

To obtain the slow eigenmodes, we match the first-order terms of Eq. (6) and substitute $\mathbf{u}(0) = \mathbf{M\tilde{u}}$, so that

$$\mathbf{Q}_{\mathcal{P}}(0)\mathbf{u}' + \mathbf{Q}'_{\mathcal{P}}\mathbf{M\tilde{u}} = \lambda'\mathbf{M\tilde{u}}. \qquad (8)$$

Multiplying from the left by the transpose of the membership matrix $\mathbf{M}$ we get

$$\mathbf{M}^T\mathbf{Q}_{\mathcal{P}}(0)\mathbf{u}' + \mathbf{M}^T\mathbf{Q}'_{\mathcal{P}}\mathbf{M\tilde{u}} = \lambda'\mathbf{M}^T\mathbf{M\tilde{u}}. \qquad (9)$$

The $i$th row of $\mathbf{M}^T$ is the trivial eigenvector $\mathbf{u}_i(w=0)$ corresponding to physical node $i$; therefore $\mathbf{M}^T\mathbf{Q}_{\mathcal{P}}(0)$ is all zeros and $\mathbf{M}^T\mathbf{M}$ is the $N \times N$ identity matrix, leading to Eq. (7) in the text.

## The fruit fly connectome

We study the Hemibrain data set, which describes a portion of the central brain of the fruit fly, *Drosophila melanogaster*[15]. The physical layout of the connectome is provided by the detailed three-dimensional shape of each neuron and the location of the synapses between them. The corresponding combinatorial network contains 21,662 nodes representing neurons and 13,603,750 links representing synapses. Synaptic partners are connected through approximately 5 synapses on average, and the maximum number of synapses between two neurons is 6039. In our calculations, we treat the combinatorial network as a weighted and undirected network, where the weight of the link $(i,j)$ is equal to the number of synapses between neurons $i$ and $j$. Note that we only require the combinatorial network and the volume of each node for our calculations; therefore, the detailed physical layout of the connectome is, in fact, not needed.

Note that the Hemibrain data set covers a large portion of, but not the entire, fruit fly brain. Since degree and volume are local properties of the nodes, we expect that the results presented here would not change significantly if the entire connectome were to be considered.

**Degree distribution.** We find that the weighted degree distribution has a heavy tail, which can be approximated by a power law with $\gamma_{ff} \approx 2.3$ for degrees $\geq 1058$ with an exponential cutoff; the power law fit, however, cannot be distinguished from a lognormal fit on the same range[43,44].

**Laplacian spectrum.** Comparing the spectrum of the combinatorial Laplacian $\mathbf{Q}_{\mathcal{G}}$ and the volume-normalized Laplacian $\mathbf{Q}_{\text{phys}} = \mathbf{V}^{-1/2}\mathbf{Q}_{\mathcal{G}}\mathbf{V}^{-1/2}$ carries a level of ambiguity: $\mathbf{Q}_{\mathcal{G}}$ does not depend on the node volumes, while changing the unit of volume multiplies the spectrum of $\mathbf{Q}_{\text{phys}}$ by a constant. To meaningfully compare the two spectra, (i) we think of $\mathbf{Q}_{\mathcal{G}}$ as a physical Laplacian where all nodes have unit volume, and (ii) we set the mean node volume in $\mathbf{Q}_{\text{phys}}$ to unity, i.e., $\langle v \rangle = 1$. With this choice of units, any difference in the eigenvalues is due to the heterogeneous distribution of node volumes in the physical network and not to a global shift caused by the choice of units.

## Reporting summary

Further information on research design is available in the Nature Portfolio Reporting Summary linked to this article.

## Data availability

Data to reproduce the figures is available at https://github.com/posfaim/physnets_as_net-o-nets.

## Code availability

Code to generate random networks and reproduce the figures is available at https://github.com/posfaim/physnets_as_net-o-nets[51].

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

## Acknowledgements

I.B., M.P., and Á.T. were funded by ERC grant No. 810115-DYNASNET. ÁT and SÖS acknowledge partial support from the Icelandic Research Fund, grant No. 239736-051. GP was funded by the ERC Consolidator Grant No. 772466-NOISE.

## Author contributions

M.P. developed and performed the numerical simulations. M.P. and I.B. performed the data analysis. G.P., Á.T., S.Ö.S., I.B., and M.P. contributed to the analytical results and the conceptual design of the study. M.P. was the lead writer of the manuscript.

## Funding

## Competing interests

The authors declare no competing interests.
