## [Peer Review File · Nature Communications]

REVIEWER COMMENTS

Reviewer #1 (Remarks to the Author):

I have reviewed the manuscript entitled “A network-of-networks model for physical networks,” written by Gabor Pete et al. This paper introduces a network-of-network framework to model the growth of physical networks whose nodes and links are usually embedded in a geometric space. Usually, the physical networks with space constraints show homogenous degree distributions, such as brain networks, road transportation networks, power grids, and many more. The authors developed a network-of-network representation of physics networks and found that the super network follows a power-law degree distribution. Furthermore, the authors derived the degree distribution and the volume of each subnetwork. Overall, the paper is well-written, and the research is novel. Yet, I have several concerns and comments on the current form of the manuscript regarding the models, methods, and results of this work. I am open to looking at a revised version if the author can satisfactorily address the issues discussed below:

1. Why this model? How can this model describe the growth of the brain network? The authors need to give some more examples or real-world data to demonstrate the rationality of this model. I believe the theoretical analysis is correct. However, I need more evidence to support why this is a suitable model.

2. Related to my first comment, I think the results about the fruit fly brain network are significant. However, the authors should have produced more results about this dataset. The authors may consider putting this figure earlier and conducting more analysis about this dataset to support this research. It would be better if the authors could find more examples/datasets to support this model.

3. The authors used LERWs to generate nodes. It needs to be clarified why using this approach. How do the random walk methods affect the results? Since the authors derived the theoretical results before the random walk method, indicating that the analysis will always hold regardless of random walk methods. Considering an extreme case that each agent only moves in one direction, the super network may be close to a chain-like network, far from Scale-Free networks as the authors predicted. Consequently, if each agent walks in one direction more than the other, the degree distribution should be more homogenous. It seems that the supernetwork may show different degree distributions depending on the random walk methods.

4. Like 3, how do the random walk methods affect the results of the Laplacian?

Reviewer #2 (Remarks to the Author):

In this manuscript, Pete et al attempt to understand the effects of embedding a network into a finite dimensional space. This problem has been treated before, but the authors introduce a Network of Networks approach which is innovative. Within this approach, they are able to demonstrate that the physical structure of the underlying space has important constraints to the dynamics and functioning of the networks.

This includes a novel degree - volume correlation which have not been taken into account before.

I like the modeling and predictive nature of the approach. I believe that the mathematical results of the paper are sound and interesting. What I am not fully convinced is on the connection of the modeling with a real-world problem. I think that the authors have not addressed in full detail the case study using the brain connectome of the fruit fly and its significance for the understanding of the brain structure and functioning. Without a clear application of the proposed method, the appeal of the approach to a broader audience is somehow diminished.

Reviewer #3 (Remarks to the Author):

In this paper, the authors propose a model for the growth of physical networks. This model is analytically tractable and has some interesting properties. Then they compare the model against the recently published reconstruction of the brain connectome of the larva of *D. melanogaster*.

I think that the manuscript, while of interest and of merit, is not above the bar for Nature Communications. The concern is not technical, it is the lack of explanatory power of the model which I think is a must for a journal of this level. As the authors point out, physical networks have received much attention in top journals in recent years, and the brain is a fantastic example of that. However, the 'physical' mechanism by which the brain grows has nothing to do with the mechanism proposed in the manuscript. As the brain in *D. mel* adds new neurons these are part of a 'family' of neurons that are basically programmed to make connections on a certain area of the brain, as a result they grow in bundles, that is, in a directed fashion guided by the expression of different receptors. That being said, it is true that there are many neuron families some of which only have one neuron. Because these neurons were in the brain from the start of development they are bigger

and have more connections as a general rule, which is aligned with some of the ingredients of the authors' physical model.

Nonetheless, the authors should note that initially the volume is small occupied with a small fraction of neurons and as more neurons are added, to make room for them the old neurons deform their shape to accommodate growth. So it is not quite the image of a space that is available and needs to be filled up, but a deformation process until it reaches the final volume.

Obviously, I understand the need for models that can help us understand general mechanisms that account for the properties we see in physical networks, and this model is a nice first step in that direction. However, these models cannot exist detached from knowledge behind the processes by which these networks actually grow, which in the case of the larval brain has been studied. If we do not do this, we run the danger of associating properties to the wrong mechanism. I really encourage the authors to get a better understanding of known processes and their proposed mechanism and try to explain when one can be effectively mapped into the other and when it cannot.

Also, it would have been nice to see if their findings are the same for other nervous systems for nematodes (not sure if they are too small for this analysis?), or the adult brain of *D melanogaster* (released end of June; also the hemibrain released a few years back).

It would also be nice to consider other physical networks that maybe fit better the physical model the authors propose. Also it would give us an idea of whether similar macro properties can be assigned to the same micro mechanism or not.

Other comments:

Some parts of the description of the model would benefit from a rewriting:

Page 2 - discussion on 'As a consequence, for saturated networks, the physical layout $P(\dots)$ ' - is confusing. I am not sure what is the point the authors are trying to make.

Page 2 (last par): 'The time derivative of V_t shows that ...' -> To obtain v_t the authors 'take a continuous time approach' and integrate eq for v_t . Therefore, it is not that the time derivative of V_t is v_t , this is the assumption they are making. - I would rewrite this part to make clear what is assumed by the model and what is not.

Other comments:

Do the authors consider the left hemisphere, the right hemisphere or both? The two hemispheres are supposed to be mirror copies of one another. I am not sure redundancy affects this analysis or not (I think it does not, but it is good to take it into account).

Reviewer #1

I have reviewed the manuscript entitled “A network-of-networks model for physical networks,” written by Gabor Pete et al. This paper introduces a network-of-network framework to model the growth of physical networks whose nodes and links are usually embedded in a geometric space. Usually, the physical networks with space constraints show homogenous degree distributions, such as brain networks, road transportation networks, power grids, and many more. The authors developed a network-of-network representation of physics networks and found that the super network follows a power-law degree distribution. Furthermore, the authors derived the degree distribution and the volume of each subnetwork. Overall, the paper is well-written, and the research is novel. Yet, I have several concerns and comments on the current form of the manuscript regarding the models, methods, and results of this work. I am open to looking at a revised version if the author can satisfactorily address the issues discussed below:

We thank the Reviewer for the positive assessment of our work, and we appreciate the constructive comments about the applications and the generality of the network growth model. We have made several major changes in response to the Referee’s suggestions, like adding a new section expanding the analysis of the fruit fly brain, re-writing the discussion section, and including additional real data sets and additional simulation results in a new Supplementary Information.

1. Why this model? How can this model describe the growth of the brain network? The authors need to give some more examples or real-world data to demonstrate the rationality of this model. I believe the theoretical analysis is correct. However, I need more evidence to support why this is a suitable model.

The Reviewer raises an important question about our motivations; we now realize that we were not clear enough about the overarching objective of our work. The goal of the network-of-networks representation is not to model the growth of brain networks specifically but to provide a general and powerful mathematical representation of complex physical networks. Our key results are the following:

- We introduce a network-of-networks representation for physical networks, a flexible tool that allows both analytical and computationally efficient description of real and model physical networks.
- Leveraging on this representation, we study an analytically solvable model of physical networks, identifying volume-degree correlations as important emergent properties that can arise due to volume exclusion.
- Building on results from multi-layer networks, we derive the physical Laplacian, which demonstrates that the volume-degree correlations not only affect the growth of networks, but also affect their dynamical properties.
- Finally, we apply the network-of-networks representation to calculate volume-degree correlations and the physical Laplacian of a large, real physical network in the main text, and four more data sets in the new Supplementary Information. We find that the volume-degree correlations are prevalent in real physical networks and affect their Laplacian spectrum analog to the model networks. The fact that volume-degree correlations emerge in a minimal model while also prevalent in real systems suggests a general mechanism behind such correlations that does not depend on the complex details of the growth of real networks.

We focus on a neural network as an example in the main text simply because among physical networks neural networks have the largest and most detailed three-dimensional maps available. Besides characterizing the fruit fly brain network, this application further highlights the computational efficiency of our methodology: recent work [1-2] on physical networks, in fact, had to narrow down their applications to single brain regions of the fruit fly brain, due to computational limitations. In contrast, we analyze the entire network without any special resources.

To further strengthen the generality of our results, we now characterize in the SI four additional data sets, namely a neural network of a nematode, a network describing the cavities of a porous material, a river network and a vascular network.

In summary, prompted by the Reviewer's comment, we made the following changes:

- (i) We re-titled the manuscript "Physical networks as network-of-networks" to better reflect its main focus.
- (ii) We adjusted to the abstract and the introduction and we re-wrote the discussion to better explain the goals of our work.
- (iii) We explain the rationale for focusing on the fruit fly brain as a test case.
- (iv) We study additional real networks to demonstrate the wide applicability of our framework and the generality of our results.

[1] Liu, Yanchen, Nima Dehmamy, and Albert-László Barabási. "Isotopy and energy of physical networks." *Nature Physics* 17.2 (2021): 216-222.

[2] Pósfai, Márton, et al. "Impact of physicality on network structure." *Nature Physics* 20.1 (2024): 142-149.

2. Related to my first comment, I think the results about the fruit fly brain network are significant. However, the authors should have produced more results about this dataset. The authors may consider putting this figure earlier and conducting more analysis about this dataset to support this research. It would be better if the authors could find more examples/datasets to support this model.

Prompted by the Reviewer's suggestion, we dedicate a separate section for the fruit fly brain in the revised manuscript. We improve and expand the analysis of the data set in three ways:

- (i) Show the weighted degree distribution (number of synapses) in Fig. 4b and Fig.4c instead of the degree distribution (number of unique synaptic partners) we showed before. This is more in line with the Laplacian spectra of Fig. 4d, where link weights were included. By taking the weights into account we found that the degree exponent is <3 and the degree-volume correlations are close to linear, meaning that the empirical results better match the model.
- (ii) We now also study the structure of the leading eigenvector of the combinatorial and physical Laplacian (Fig. 4e-g). We find that in both cases the eigenvectors are concentrated on a single node: for the combinatorial network, as expected, the central node is the largest hub; in contrast, for the physical Laplacian the central node is the node that has the highest degree-volume ratio, i.e., it balances the high degree with low node volume.
- (iii) To further support the generality of our framework and results, we now include four additional real networks to the newly added Supplementary Information: a vascular network, a rock pore network, a river network and the neural network of the *C. elegans*. In each case, we find a positive

degree-volume correlation, which affects the Laplacian spectrum analog to the model networks and the fruit fly brain.

3. The authors used LERWs to generate nodes. It needs to be clarified why using this approach. How do the random walk methods affect the results? Since the authors derived the theoretical results before the random walk method, indicating that the analysis will always hold regardless of random walk methods. Considering an extreme case that each agent only moves in one direction, the super network may be close to a chain-like network, far from Scale-Free networks as the authors predicted. Consequently, if each agent walks in one direction more than the other, the degree distribution should be more homogenous. It seems that the supernetwork may show different degree distributions depending on the random walk methods.

Indeed, as the Reviewer points out, the analysis is not specific to LERWs and holds under some mild assumptions on the random walk (or more general randomly growing pieces). Besides the knowledge of the fractal dimension of the random walk, we need it to have the property that if the boxes around two random walk pieces intersect, then with uniformly positive probability the pieces also intersect (resulting in Eq. (1) in the manuscript). This holds for many examples but does not hold for extreme cases like a walk in 2 dimensions that only steps in the horizontal direction. More generally, we require a level of isotropy for the node growth: nodes don't favor certain directions so much as to avoid each other. If the above assumption is met, our model produces a degree distribution with a power law tail independent of the type of random walk used; the exponent of the power law, however, does depend on the fractal dimension of the random walk trajectories. Our choice to focus on LERWs as an example has two main reasons: first, LERWs are well-studied with known non-trivial fractal exponents, and second, LERWs provide a simple method to produce self-avoiding trajectories that do not trap themselves (for example, the kinetic version of the self-avoiding walk traps itself in 2 and 3 dimensions). In the newly added Supplementary Information, we provide additional simulations using various random trajectories (simple random walk, kinetic self-avoiding random walk, random jumps and random rays) to show the validity of our analysis beyond LERWs.

In summary, in response to the Reviewer's comment:

- (i) We clarify the assumptions about the random walk in our analytical derivation. We use the example proposed by the Reviewer as an example for the necessity for these assumptions.
- (ii) We compare the analytical prediction to additional random walks in the Supplementary Information accompanying our revised manuscript.

4. Like 3, how do the random walk methods affect the results of the Laplacian?

The spectrum of the physical Laplacian is affected by the power law degree distribution of the super-network and the degree-volume correlations. Both features emerge independently of the type of random walk used (under the mild assumptions discussed above); therefore, the spectrum of the physical Laplacian is also affected similarly. We demonstrate the general emergence of positive degree-volume correlations through extensive simulations using various random walks in the newly added Supplementary Information.

In addition, we provide arguments why positive degree-volume correlations and the suppression of the tail of the Laplacian spectrum are expected to arise beyond our model, through any alternative network generation process that minimizes the total network volume. Any combinatorial network G has many possible physical realizations P , a minimum volume realization is a P that minimizes the total volume of the network. Consider a node in G with degree k ; in any possible P , the physical realization of the node must have volume at least proportional to k , otherwise it is unable to support k connections. This means that any physical network generation process that somehow minimizes total volume is characterized by positive degree-volume correlations. Volume minimization can happen either explicitly or as an emergent property, like in our model: either a node must be large because it needs to create many connections or has many because it is large. Because of these general correlations, we expect that for networks that at least to some extent minimize volume, no matter how complex the generation process is, the tail of the physical Laplacian is suppressed by the large volume of hubs. We now clarified this argument at the end of page 5 of the revised manuscript.

The question of the Reviewer, however, also points towards a more general open question. The derivation of the physical Laplacian shows that physical networks with the same combinatorial graph, but different physical layout can have very different spectra. To what extent is the spectrum determined by the combinatorial graph? Can we design a physical layout for a given combinatorial network to control the spectrum? We now mention these open questions in the discussion of the revised manuscript.

Reviewer #2

In this manuscript, Pete et al attempt to understand the effects of embedding a network into a finite dimensional space. This problem has been treated before, but the authors introduce a Network of Networks approach which is innovative. Within this approach, they are able to demonstrate that the physical structure of the underlying space has important constraints to the dynamics and functioning of the networks.

This includes a novel degree - volume correlation which have not been taken into account before.

I like the modeling and predictive nature of the approach. I believe that the mathematical results of the paper are sound and interesting.

We appreciate that the Reviewer found our modelling and mathematical results compelling. We also thank the Reviewer for appreciating the novelty of our characterization of degree-volume correlations. We address in detail the constructive comments of the Reviewer below.

What I am not fully convinced is on the connection of the modeling with a real-world problem.

The Reviewer raises an important point about the applicability of the network-of-networks framework to real physical networks. We realize that in the manuscript we need to better describe the advantages of the framework with respect to previous approaches and to further illustrate the possibilities it affords. As mentioned above by the Reviewer, embedding networks that obey volume exclusion have been studied before, for example, in the recent Refs. [1-3], which provide great insights into the mechanisms shaping physical networks. Our work pertaining to real networks goes beyond the existing literature in two important ways:

- (i) In these references, simple models of generating physical networks are introduced that represent physical nodes as spheres and links as tubes. Such representation is a natural first step; however, it comes with severe limitations: it is unable to capture the complex shape of real nodes, one must limit the maximum degree of nodes or allow links to violate volume exclusion in the vicinity of nodes.

Therefore to compare models to real networks, the models must rely on a representation that is able to describe arbitrary node shapes. One such representation is provided by the network-of-networks framework. This also connects the analysis of physical networks to the rich literature of multilayer networks, opening new avenues of research.

- (ii) The methods introduced in Refs. [1-3] require the full shape of real physical networks as input, hence are computationally expensive to measure. For this reason, their applications are limited to real networks with a few hundred nodes or require extensive computational resources. On the other hand, in our work, we used to the network-of-networks representation and the network growth model to identify node degree-volume correlations and the physical Laplacian as important descriptors of the structure and dynamics on physical networks. Both of which, requiring only the combinatorial network and the node volumes as input, are cheap to compute. This is well illustrated by the case study of the fruit fly brain: Refs. [2] and [3] also analyze the

same data set, however, they restrict their applications to single brain regions. In contrast, we characterize the entire network.

We also remark that the model of network growth introduced in the manuscript is not intended to model any specific real system. Our goal was to devise a minimal model that captures volume exclusion and complex node shapes. The fact that we observe volume-degree correlations in real systems and that it also emerges in a minimal model implies that these correlations do not depend on the specific details of the systems, rather there is a general mechanism behind it.

In response to the Reviewer's comment, we made the following changes to the manuscript:

- (i) We now include several additional real networks (a vascular network, a rock pore network, a river network and the *C. elegans* neural network) in the newly added Supplementary Information to demonstrate the wide applicability of our framework and to further support the generality of our results.
- (ii) We extended the analysis of the fruit fly brain network in the revised manuscript: we now study the structure of the leading eigenvector of the physical Laplacian and the role of hubs (see details in our answer to the next comment).
- (iii) We expand on the conceptual and computational advantages of the network-of-networks framework and our methods in the discussion of the revised manuscript.

[1] Dehmamy, Nima, Soodabeh Milanlouei, and Albert-László Barabási. "A structural transition in physical networks." *Nature* 563.7733 (2018): 676-680.

[2] Liu, Yanchen, Nima Dehmamy, and Albert-László Barabási. "Isotopy and energy of physical networks." *Nature Physics* 17.2 (2021): 216-222.

[3] Pósfai, Márton, et al. "Impact of physicality on network structure." *Nature Physics* 20.1 (2024): 142-149.

I think that the authors have not addressed in full detail the case study using the brain connectome of the fruit fly and its significance for the understanding of the brain structure and functioning. Without a clear application of the proposed method, the appeal of the approach to a broader audience is somehow diminished.

We thank the Reviewer for pointing out this shortcoming in the original version of our manuscript. We used the case study of the fruit fly brain to demonstrate the applicability of our framework to large-scale real networks and that heterogeneous degree distributions and degree-volume correlations (as suggested by our network-of-networks model) are indeed present in real systems. We derived the physical Laplacian to show that these structural properties in turn strongly affect the dynamics unfolding on physical networks. At this point, we stopped, missing the chance to explore the consequences of our findings on the dynamics and to better explain the method's applications. To remedy this, we now compare the leading eigenvector of the physical Laplacian to that of the combinatorial Laplacian. We find that the leading eigenvector in the combinatorial network is concentrated on the hubs, meaning that the hubs have an outsized influence on early spreading. On the other hand, taking node volumes into account, the leading eigenvector of the physical Laplacian becomes localized on the node that has the maximum

degree-volume ratio, balancing relatively high connectivity and low volume. (Specifically, it is the 159th largest degree node and is at the top 15 percentile of the volume distribution.) This indicates that the physical layout not only suppresses the tail of the Laplacian, but also changes the identity of the early spreaders in the network.

Our initial explorations in this direction indicate that this is part of a more general phenomenon which can be investigated by systematically tuning the node volumes. We now discuss these relevant perspectives in the conclusion of our revised manuscript, whose detailed investigation is left for future work. We are grateful to the Reviewer for prompting us to explore this direction.

In addition to the expanded analysis of the fruit fly brain network, we now include the analysis of four further real networks in Sec. S2 of the newly added Supplementary Information.

We thank the Reviewer for the valuable observations and criticisms which helped us identify and resolve crucial shortcomings in the presentation of our framework. We are confident that the additional descriptions and new results about other physical networks make now clear the broad applicability and relevance of our framework and results.

Reviewer #3

In this paper, the authors propose a model for the growth of physical networks. This model is analytically tractable and has some interesting properties. Then they compare the model against the recently published reconstruction of the brain connectome of the larva of *D. melanogaster*.

I think that the manuscript, while of interest and of merit, is not above the bar for Nature Communications. The concern is not technical, it is the lack of explanatory power of the model which I think is a must for a journal of this level. As the authors point out, physical networks have received much attention in top journals in recent years, and the brain is a fantastic example of that.

We thank the Reviewer for the positive assessment of our model and analytical results, and we appreciate the constructive comments about applying our results to the fruit fly brain connectome. We made several major changes in response to the Reviewer's suggestions, like expanding the analysis of the connectome, including additional real data sets, re-writing the discussion and changing the title of the manuscript. We also explored extensions of our model to incorporate additional growth mechanisms.

However, the 'physical' mechanism by which the brain grows has nothing to do with the mechanism proposed in the manuscript. As the brain in *D. mel* adds new neurons these are part of a 'family' of neurons that are basically programmed to make connections on a certain area of the brain, as a result they grow in bundles, that is, in a directed fashion guided by the expression of different receptors. That being said, it is true that there are many neuron families some of which only have one neuron. Because these neurons were in the brain from the start of development they are bigger and have more connections as a general rule, which is aligned with some of the ingredients of the authors' physical model.

Nonetheless, the authors should note that initially the volume is small occupied with a small fraction of neurons and as more neurons are added, to make room for them the old neurons deform their shape to accommodate growth. So it is not quite the image of a space that is available and needs to be filled up, but a deformation process until it reaches the final volume.

Obviously, I understand the need for models that can help us understand general mechanisms that account for the properties we see in physical networks, and this model is a nice first step in that direction. However, these models cannot exist detached from knowledge behind the processes by which these networks actually grow, which in the case of the larval brain has been studied. If we do not do this, we run the danger of associating properties to the wrong mechanism. I really encourage the authors to get a better understanding of known processes and their proposed mechanism and try to explain when one can be effectively mapped into the other and when it cannot.

We agree with the Referee on the model's simplicity – and it is designed to be so on purpose. To develop a mathematical theory of physical networks, we must devise the simplest models that capture the essential features of physical networks: complex node shape and volume exclusion. We chose the *D. melanogaster* connectome as a case study because it represents one of the largest and most detailed maps of physical networks currently available.

The first steps of modelling physical networks were developed in Refs. [1-2], both references represented physical nodes as spheres and links as tubes. In that respect, our work provides several significant advancements:

- (i) We introduce a network-of-networks representation for physical networks to supersede the sphere-and-tube representation. This conceptual shift allows both analytical and computationally efficient description of real and model physical networks.
- (ii) Leveraging on this representation, we study an analytically solvable model of physical networks. The model identifies volume-degree correlations as important emergent properties that naturally arise in even the simplest systems obeying volume exclusion.
- (iii) Through the derivation of the physical Laplacian, we uncover a fundamental mechanism through which structural properties emerging due to physicality affect dynamics on physical networks.

The use of the connectome of the fruit fly brain as a case study allows us to demonstrate that the network-of-networks framework is useful to describe large-scale real physical networks. In doing so, we show that the emergent properties identified by the model as important are present in the connectome as well. We do not claim, however, that this is an effect of pure volume exclusion – which we now make explicitly clear in the new version of the manuscript.

The mathematical know-how developed in our manuscript lays the groundwork for further explorations. Incorporating the mechanisms listed by the Reviewer into a network-of-networks model is conceptually straightforward, allowing numerical, and perhaps analytical, investigation of more complex models. Taking the suggestion of the Reviewer to heart, we explored several extensions to the growth model. For example, the figure below shows results for a version that incorporates additional attraction or repulsion between the random walks, guiding the growth of physical nodes. We find that repulsion increases the heterogeneity in the node degree, pushing the system towards a star-like topology, while attraction between physical nodes leaves the degree exponent unchanged. In both cases, volume-degree correlations emerge.

Network growth with attraction or repulsion. In this version of the model, we grow nodes via LERWs except if a node trajectory is adjacent to an existing node, we bias the probability to hit the node and terminate the growth. Parameter value <1 corresponds to repulsion and >1 to attraction; while with parameter value 1 we recover the original model. **(a)** The average node volume as a function of attraction/repulsion. With strong repulsion, (dashed line) nodes avoid forming connections, increasing the average node size. **(b)** The degree distribution is power law in case of attraction, for repulsion (dashed line) the initial nodes gather outsized number of connections in a winner-takes-all type dynamics. Note that the lower maximum degree for repulsion is due to less, but larger nodes being added to the lattice. **(c)** In all cases, we observe strong volume-degree correlations. Each line corresponds to a single run on a three-dimensional lattice with side length 22.

Ultimately, we decided to forego the above and other extensions in our current manuscript, as we feel it is out of scope and discussion of such specialized models would shift focus from our fundamental message: network-of-networks is a powerful framework to understand physical networks, providing a set of analytically tractable and computationally efficient tools.

Lastly, we would like to remark that although our minimal model led us to discover the effect of positive degree-volume correlations on the Laplacian spectrum, we have strong arguments that such features arise under more complex physical network generation processes. Any combinatorial network G has many possible physical realizations P , a minimum volume realization is a P that minimizes the total volume of the network. Consider a node in G with degree k ; in any possible P , the physical realization of the node must have volume at least proportional to k , otherwise it is unable to support k connections. This means that any physical network generation process that somehow minimizes total volume is characterized by positive degree-volume correlations. Volume minimization can happen either explicitly or as an emergent property, like in our model: either a node must be large because it needs to create many connections or has many because it is large. Because of these general correlations, we expect that for networks that at least to some extent minimize volume, no matter how complex the generation process is, the tail of the physical Laplacian is suppressed by the large volume of hubs. We now clarified this argument at the end of page 5 of the revised manuscript.

We do realize that we did not fully explain the rationale of the framework. Prompted by the Referee's legitimate comments, we made the following changes:

- (i) To better explain our aims, we changed the title, extended the abstract and introduction, and re-wrote the discussion.
- (ii) We made explicit the scope of the growth model and the reasons for using the connectome as our main case study.
- (iii) We clarified our arguments about emergent degree-volume correlations under more complex network generation processes at the end of page 5.
- (iv) We discussed several possible generalizations of the growth model in the Discussion.

[1] Dehmamy, Nima, Soodabeh Milanlouei, and Albert-László Barabási. "A structural transition in physical networks." *Nature* 563.7733 (2018): 676-680.

[2] Pósfai, Márton, et al. "Impact of physicality on network structure." *Nature Physics* 20.1 (2024): 142-149.

Also, it would have been nice to see if their findings are the same for other nervous systems for nematodes (not sure if they are too small for this analysis?), or the adult brain of *D. melanogaster* (released end of June; also the hemibrain released a few years back). It would also be nice to consider other physical networks that maybe fit better the physical model the authors propose. Also it would give us an idea of whether similar macro properties can be assigned to the same micro mechanism or not.

Following the Reviewer's suggestion, we now include several additional real systems to the Supplementary Information accompanying our revised manuscript, including a data set representing the nervous system of the *C. elegans*. In all real systems, we find positive node degree-volume correlations. We also show that these correlations affect dynamics unfolding on the network in a similar way to the fruit fly connectome and the model networks: hubs have large volume, which counteracts the effect of their many connections. The fact that these properties are prevalent in real systems and also emerge in the simplest models suggests that they are caused by general mechanisms and do not depend on the details of the systems.

We further remark that the data for the connectome of the *D. melanogaster* studied in the main text correspond to the Hemibrain data (not the recent release of the larvae and the adult connectome). We now explicitly state this in the revised manuscript.

Other comments:

Some parts of the description of the model would benefit from a rewriting:

Page 2 - discussion on 'As a consequence, for saturated networks, the physical layout $P(\dots)$ - is confusing. I am not sure what is the point the authors are trying to make.

We understand that we were unclear about the optimality of the embeddings produced by the growth model. The point that we tried to make is that in any physical representation of a combinatorial network each node has a non-zero volume, which provides a somewhat trivial lower bound: the total volume of the network is at least $\sim N$ in the large network limit (where N is the number of physical nodes). This lower bound, however, is not always achievable: a fully connected network where all nodes are connected to all other nodes requires at least $\sim N^2$ volume. Our statement is that the physical layouts produced by our model are in fact optimal in this sense, i.e., the average node volume remains constant in the large N limit. We find this noteworthy since the growth of the network and the nodes are both random without any optimization of the embedding volume.

We clarify our statement in the revised manuscript as:

"Equation (3) predicts that N_{sat} , the number of nodes when the network saturates, scales as $N_{\text{sat}} \sim L^d$, meaning that the average node volume $\langle v \rangle$ remains constant in the $L \rightarrow \infty$ large system limit. Therefore, the physical layout P is optimal in the sense that no physical representation of a combinatorial network of N_{sat} nodes can fit into a smaller volume than $\sim N_{\text{sat}} \sim L^d$. It is noteworthy that the model achieves this bound despite the fact that the nodes grow randomly."

Page 2 (last par): ‘The time derivative of V_t shows that ...’ -> To obtain v_t the authors ‘take a continuous time approach’ and integrate eq for v_t . Therefore, it is not that the time derivative of V_t is v_t , this is the assumption they are making. - I would rewrite this part to make clear what is assumed by the model and what is not.

We re-wrote the sentence in question to clarify the approximation as

“In the continuous time approximation, the volume of the newly added node v_t is provided by the time derivative of V_t , i.e., $v_t \sim \left(t/L^d\right)^{-d_{Tf}/d}$.”

Other comments:

Do the authors consider the left hemisphere, the right hemisphere or both? The two hemispheres are supposed to be mirror copies of one another. i am not sure redundancy affects this analysis or not (I think it does not, but it is good to take it into account).

In our manuscript we use the entire Hemibrain dataset, which does not contain both hemispheres completely, but there is some redundancy [1]. This does not affect most properties that we investigate: volume-degree correlations are local properties of the nodes, including the entire network would not significantly affect the observed correlations. The suppression of the tail of the Laplacian depends on the presence of strong degree-volume correlations; therefore, we do not expect a change in the overall behavior of the spectrum.

In the revised manuscript, we extended the analysis of the connectome by studying the structure of the leading eigenvectors. Through this analysis, we identified the nodes that are the fastest spreaders in the network. Including the entire connectome might change the identity of these early spreaders; however, the nodes that we identify in the limited data set will remain one of the most important ones.

We now include the following remark about the limitations of the data set in the Methods and materials section:

“Note that the Hemibrain data set covers a large portion of, but not the entire, fruit fly brain. Since degree and volume are local properties of the nodes, we expect that the results presented here would not change significantly if the entire connectome would be considered.”

[1] <https://www.janelia.org/project-team/flyem/hemibrain>

We thank the Reviewer for their valuable comments which have motivated us to delve deeper into crucial aspects of our work. We feel this has considerably improved the quality of our presentation, emphasizing the generality and the broad impact of our results.

REVIEWERS' COMMENTS

Reviewer #1 (Remarks to the Author):

The authors have addressed all my comments, and I would like to accept this manuscript for publication in Nature Communications.

Reviewer #2 (Remarks to the Author):

The authors have satisfactorily addressed my concerns.

Reviewer #3 (Remarks to the Author):

I think the authors have made a great effort to reformulate their manuscript in a way that is more general. They have addressed all of my comments (and I think those of the other reviewers) quite convincingly. I think that the degree-volume correlation is a very nice result, that seems to hold for several of the physical networks the authors look at.

I congratulate them for their effort of making a new round of analysis and for providing careful answers to all of the comments.